# Monitoring People's Emotions and Symptoms from Arabic Tweets during the COVID-19 Pandemic

Ali Al-Laith [1,*] and Mamdouh Alenezi [2]

1   Center for Language Engineering, Al-Khawarizmi Institute of Computer Science, University of Engineering and Technology, Lahore 54890, Pakistan
2   College of Computer and Information Sciences, Prince Sultan University, Riyadh 11586, Saudi Arabia; malenezi@psu.edu.sa
*   Correspondence: ali.allaith@kics.edu.pk

**Abstract:** Coronavirus-19 (COVID-19) started from Wuhan, China, in late December 2019. It swept most of the world's countries with confirmed cases and deaths. The World Health Organization (WHO) declared the virus a pandemic on 11 March 2020 due to its widespread transmission. A public health crisis was declared in specific regions and nation-wide by governments all around the world. Citizens have gone through a wide range of emotions, such as fear of shortage of food, anger at the performance of governments and health authorities in facing the virus, sadness over the deaths of friends or relatives, etc. We present a monitoring system of citizens' concerns using emotion detection in Twitter data. We also track public emotions and link these emotions with COVID-19 symptoms. We aim to show the effect of emotion monitoring on improving people's daily health behavior and reduce the spread of negative emotions that affect the mental health of citizens. We collected and annotated 5.5 million tweets in the period from January to August 2020. A hybrid approach combined rule-based and neural network techniques to annotate the collected tweets. The rule-based technique was used to classify 300,000 tweets relying on Arabic emotion and COVID-19 symptom lexicons while the neural network was used to expand the sample tweets that were annotated using the rule-based technique. We used long short-term memory (LSTM) deep learning to classify all of the tweets into six emotion classes and two types (symptom and non-symptom tweets). The monitoring system shows that most of the tweets were posted in March 2020. The anger and fear emotions have the highest number of tweets and user interactions after the joy emotion. The results of user interaction monitoring show that people use likes and replies to interact with non-symptom tweets while they use re-tweets to propagate tweets that mention any of COVID-19 symptoms. Our study should help governments and decision-makers to dispel people's fears and discover new symptoms associated with the symptoms that were declared by the WHO. It can also help in the understanding of people's mental and emotional issues to address them before the impact of disease anxiety becomes harmful in itself.

**Keywords:** emotion detection; COVID-19; COVID-19 symptoms; text classification; arabic tweets

## 1. Introduction

The COVID-19 disease has been declared a pandemic due to the quick spread of the virus all over the world. It profoundly affects all aspects of society, including mental and physical health. COVID-19's rapid spread created a global public health crisis that made governments limit some activities including the non-essential economy and social activities, and there were disruptions such as border closing, loss of jobs, airline shutdowns, and financial breakdowns. These insecurities and disturbances cause citizens to feel anxious, fearful, and depressed. In any public health crisis, tracking and monitoring systems helps the concerned decision-makers to create situational awareness of disease spread and to identify newly affected areas to initiate appropriate and timely responses. The

damage and direct impact of a crisis, such as economic losses [1], hospitalized patients [2], and death rates [3] are often monitored. However, there are few studies or systems that provide awareness of citizens' feelings towards the pandemic, especially for low-resourced languages such as Arabic [4–7].

Infodemiology is the science that uses the information in an electronic medium such as social media platforms to help public health and policy-makers assess and forecast epidemics and outbreaks [8]. It has been used in different applications related to the COVID-19 pandemic, including the tracking of the spread of the new COVID-19 disease [8], monitoring of the COVID-19 trends in Google Trends [9], determine the correlation between internet search for some symptoms and the confirmed case count of the COVID-19 [10], and investigating the information about the prevention of COVID-19 on the internet [11], etc.

Twitter, a social media platform, is one of the popular platforms used to conduct social media research in academia and industry [12]. It is used as a data source in many applications, including the monitoring of mental health [13], disease [14], transportation [15], patient safety [16], and customer opinions [17], etc. The ease of obtaining data from Twitter through its APIs attracts more researchers to conduct their research on Twitter's data.

To understand the psychological and psychological implications of a pandemic, the emotions involved, such as fear and anger, must be taken into account. In this research, we used Twitter data to analyze the emotional reactions of citizens during the COVID-19 pandemic. Concerns may also be raised due to uncertainty, political changes, and the contradictory actions of different sectors of government. We analyzed six types of emotion to find citizens' concerns to create situational awareness of emotional stress and anxiety levels in healthy individuals and intensify the symptoms of those with pre-existing psychiatric disorders. We have classified the dataset into six different emotions. These emotions have been identified by the psychologist Paul Echman as being universally experienced in all human cultures as shown in (https://www.verywellmind.com/an-overview-of-the-types-of-emotions-4163976 (accessed on 18 February 2021)). We also compare these types of emotion with tweets that mention any of the disease symptoms, to study citizens' fears and analyze them by doctors and psychiatrists.

The majority of existing monitoring systems were developed for languages such as English and other European languages. To the best of our knowledge, this is the first study in Arabic to monitor people's emotions during the COVID-19 pandemic from textual data on social media. The major contributions of this research are as follows:

- The building and annotating of large Arabic emotion and symptom corpora from Twitter.
- Developing a system for monitoring people's emotions and link these emotions with tweets that mention any of the COVID-19 pandemic symptoms.

The rest of the paper is organized as follows: In Section 2, we present a literature survey of the existing monitoring systems and discuss their approaches and performance. In Section 3, we present our methodology for the corpus generation process and show in detail the data collection, emotional annotation, statistical information about the corpus, and deep learning classifier. In Section 4, we present the experimental results and analysis of the rule-based and automatic annotation on our corpus and the emotion monitoring over time, including the evolution and spikes of emotions, tracking emotions over time, and tracking user interaction. In Section 5, we present a discussion about the proposed emotion dataset and monitoring system. In Section 6, we conclude with a discussion of our work and present our future plan.

## 2. Related Work

Emotion detection is one of the natural language processing and text analytics fields used to discover people's feelings in written texts [18]. It has been applied in different tracking systems, including disasters and social media monitoring. In this section, we will show the latest research on emotion monitoring and tracking with the main focus on the extraction of emotion from Arabic text and COVID-19.

Tracking citizens' concerns during the COVID-19 pandemic has been studied in [19]. The authors presented an approach to measure and track citizens' concern levels by analyzing the sentiment in Twitter data using the ratio of negative and very negative tweet counts over the total number of tweets on the dataset. They studied 30,000 tweets from March 14, 2020 to show the degree of concern by US states. Medical news and network analysis on Twitter data in Korea have been proposed in [20]. The study investigated information transmission networks and news sharing related to COVID-19 tweets. The study aimed to show how COVID-19 issues circulated on Twitter through network analysis. It has also shown that monitoring public conversations and spreading media news can assist public health professionals to make fast decisions. Tracking mental health and symptom mentions on Twitter during the COVID-19 pandemic has been studied in [21]. The study showed that social media can be used to measure the mental health of a country during a public health crisis. It can also enable the early detection of disease symptoms. A real-time dashboard ( https://bit.ly/penncovidmap (accessed on 18 February 2021)) was developed to monitor the change in anxiety, top symptom mentions on Twitter in the US, and the common and most discussed topics related to healthcare. Emotion analysis using English Twitter data during the COVID-19 pandemic was presented in [22]. They used the National Research Council Canada (NRC) Word–Emotion Association Lexicon to classify the collected data into eight basic emotion categories (anger, fear, disgust, anticipation, joy, sadness, surprise, and trust). By using emotion analysis, authorities can better understand the mental health of the people.

Arabic Twitter data was used to study depressive emotions [23]. They collected data from the Gulf region targeting people who self-declared the diagnosis of depression. They collected another set of data as a control group to be labeled as non-depressed. They later built a predictive model using different supervised learning algorithms to see if the user's tweet is depressed or not. Their feature set includes symptoms of clinical depression and online depression-related behavior. The data was unbalanced since they have 27 depressed people and 62 non-depressed people. They used the classical metrics for measuring the performance of their model (precision, recall, f-measure, and accuracy).

In [24], the authors highlighted the importance of looking after the mental health and psychosocial crisis that caused the COVID-19 outbreak. They argued that mental health and wellbeing are essential parts of healthcare; consequently, studying and mitigating these issues is vital for a stable and healthy community. They also explored some factors that might contribute to mental health issues: the uncertainty of this new illness; the unpredictability of new risks including self-isolation, social distancing, and quarantine; impaired social functioning; interpersonal issues; the perpetuation of emotional and behavioral disorders and psychological problems; predisposed mental health issues; and the tendency of being easily affected by traumatic events.

The magnitude of the novel corona virus (COVID-19) pandemic has led to considerable economic hardships, stress, anxiety, and concerns about the future. Social media can provide a place for measuring the pulse of mental health in communities. Social media plays a vital role in recording the reactions, opinions, and mental health features of social media users, as was found when the changes in psycholinguistic features before and after a lockdown in Wuhan and Lombardy were studied and the differences in the frequencies of word categories before and after lockdown were compared [25].

In [26], the study analyzed 167,073 tweets collected from the beginning of February to mid-March 2020. They studied the word frequencies and applied the Latent Dirichlet Allocation (LDA) technique to identify the most common topics in these tweets. Their analysis found 12 topics, which later on clustered into four main groups: the virus origin; its sources; its impact on people, countries, and the economy; and ways of mitigating the risk of infection.

The study of people's emotions shifting from fear to anger was presented in [27]. The authors used over 20 million tweets during the COVID-19 outbreak to study people's sentiment and its evolution. A list of topics, sentiments, and emotion attributes was used to

annotate a dataset containing 63 million tweets [28]. The study reported basic descriptive statistics about the discussed topics, sentiments, and emotions and their temporal distributions. Monitoring depression trends on Twitter during the COVID-19 pandemic has been proposed in [29]. The authors collected a large English Twitter depression dataset containing 2575 users with their past tweets. They trained three classification models to examine the effect of depression on people's Twitter language. Deep-learning model scores with psychological text features and user demographic information were combined to investigate these features with depression signals. Understanding the mindset of Indian people during the lockdown using Twitter data has been proposed in [30]. The authors used Python and R statistical software to analyze the collected dataset.

COVID-19 places more pressure on doctors and other health care workers [31]. Such pressure brings a high risk of psychological distress for doctors. To reduce mental health stigma in clinical workplaces, the study suggested supporting doctors and their families during the pandemic. An investigation of the mental health status of health staff was carried out in [32] to identify the population requiring psychological intervention. A survey was conducted to investigate three mental health issues: psychological distress, depressive symptoms, and anxious symptoms. The study recommended that the health workers in high-risk departments should receive psychiatric interventions. The health authorities have to consider setting up mental health teams for dealing with mental health problems and give psychological help and support to patients and health care workers [33]. The study suggested using web applications to monitor and evaluate the stress, anxiety, and depression levels of health care workers and provide treatment and diagnosis for them. Another research study conducted a questionnaire for 145 healthcare professionals working on COVID-19 wards in Italy [34]. The study asked healthcare workers to provide sociodemographic and clinical information in order to understand quality-of-life and mental health issues including anxiety and depression. The results show that healthcare professionals reported higher levels of depression and stress symptoms. The study suggested that healthcare workers developed mental health symptoms when working with COVID-19-infected people.

For the detection of the symptoms from Arabic text on social media, a research study has been presented in [35]. The authors used tweets in Arabic to identify the most common symptoms reported by COVID-19-infected people. The top three symptoms that were reported were fever, headaches, and anosmia. The analysis of the spread of influenza in the UAE has been proposed in [36]. The analysis was performed based on Arabic Twitter data. The authors proposed a system that can filter and extract features from the tweets and classify them into three categories: self-reporting, non-self-reporting, and non-reporting. The tweets were used to predict the number of future hospital visits using a linear regression classifier. Different topics about the spread of COVID-19 have been investigated in [37]. The study examined how Arabic people reacted to the COVID-19 pandemic on Twitter from different perspectives, including symptoms and negative emotions such as sadness.

As social media become important and ubiquitous for social networking, they support backchannel communications and allow for wide-scale interaction [38]. Social media content can be used to build tracking systems in many applications. The tracking of sentiment on news entities over time [39] is one of the systems that attract researchers in monitoring socio-politics issues. Sentiment-spike detection has been presented in [40]. The authors used Twitter data and analyzed the sentiment towards 70 entities from different domains. Tracking health trends from social media has been studied in [41]. The authors introduced an open platform that uses crowd-sourced labeling of public social media content. Tourism is another domain for which tracking systems have been built; specifically, tracking systems have been used to monitor tourist comments being written on social media [42]. The authors used a sentiment analysis lexicon to track the opinion of the tourists in Tunisia. Monitoring people's opinions before or during elections is useful to track and analyze the campaigns using Twitter data [43]. The study identified the topics that were most causal for public opinion, and show the usefulness of measuring

public sentiment during the election. Tracking and monitoring earthquake disasters from Weibo Chinese social media content was proposed in [44]. The authors focused on how to detect disasters from massive amounts of data on a micro-blogging stream. They used sentiment analysis to filter the negative messages to carry out incident discovery in a post-disaster situation.

To summarize, building a tracking system on top of social media content is very useful for governments and decision-makers. Most of the proposed work has been done for English data with fewer contributions in Arabic. The research related to the emotions and symptoms in Arabic text did not focus on the monitoring and tracking of emotions and symptom evolution over time. That motivated us to conduct our research on Arabic data to help health authorities, governments, and decision-makers to understand people's emotions during the COVID-19 pandemic.

### 3. Methodology

The architecture of the proposed monitoring system is presented in Figure 1. The proposed framework starts with data collection from Twitter. Arabic emotion lexicons were used to annotate a list of 300,000 tweets using a rule-based approach. We then use deep learning classifiers to ensure the quality of the annotated tweets. The best deep learning model was used to annotate the large unlabeled dataset. A list of COVID-19 symptom keywords was extracted and prepared to be used in symptom mentions classification. The last step was to store the annotated dataset in a database to be used to perform monitoring tasks. This work intended to build an Arabic temporal corpus that can be used for the monitoring of people's emotions and symptom mentions during the COVID-19 pandemic. To address the need, the development of the corpus has four steps: (1) data collection, (2) data preprocessing, (3) emotion annotation, and (4) symptom tweet detection.

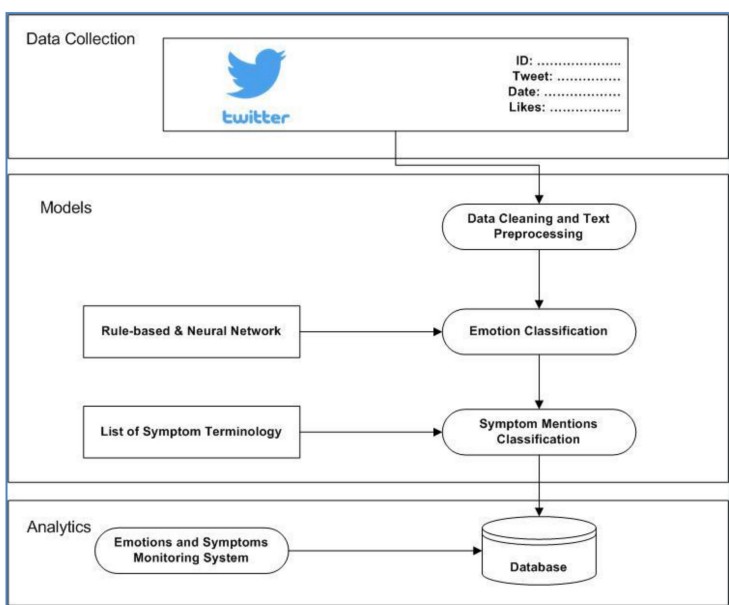

**Figure 1.** Emotions and Symptoms Monitoring System.

### 3.1. Data Collection

We used Twint, an advanced Twitter scraping tool, to collect data from Twitter and build our dataset. The tool has an option to filter tweets based on the language; therefore, we selected the Arabic language to retrieve Arabic tweets only. The collected tweets spanned the period from 1 January 2020 to 31 August 2020. To collect the relevant Twitter data, we explored the trending and most popular hashtags in Arab countries. A list of hashtags was prepared to be used as search keywords to retrieve relevant tweets from Twitter during the pandemic, as shown in Table 1. This approach focuses on getting all

of the tweets that are talking about COVID-19 in the context of Arab countries. Along with these hashtags, we used multiple queries to build joining terms related to COVID-19 with the name of all Arab world countries such as السعودة_كورونا# "#Corona_Saudia", الكوت_كورونا# "#Corona_Kuwait", etc.

**Table 1.** List of hashtags.

| Sr. # | Hashtag | Translation | Sr. # | Hashtag | Translation |
|-------|---------|-------------|-------|---------|-------------|
| 1 | #كورونا | #Coronavirus | 9 | #قطر_كورونا | #corona_Qatar |
| 2 | #المستجد_كورونا | #new_corona | 10 | #الاردن_كورونا | #corona_Jordan |
| 3 | #الجديد_كورونا | #new_corona | 11 | #السعودية_كورونا | #corona_Saudi_Arabia |
| 4 | #الصحي_الحجر | #Quarantine | 12 | #الكويت_كورونا | #corona_Kuwait |
| 5 | #المنزلي_الحجر | #Quarantine | 13 | #لبنان_كورونا | #corona_Lebanon |
| 6 | #البيت_في_خليك | #Stay_home | 14 | #كورونا البحرين | #corona_Bahrain |
| 7 | #العراق_كورونا | #corona_Iraq | 15 | #مصر_كورونا | #corona_Egypt |
| 8 | #ايران_كورونا | #corona_Iran | 16 | #اليمن_كورونا | #corona_Yemen |

The result of data collection using popular hashtags and keywords is a dataset that contains around 5.5 million unique tweets that are posted by 1.4 million users with an average of approximately four tweets per user. Table 2 shows the top 20 users in terms of the number of tweets in our dataset. The table shows that 14 accounts from the top 20 list are verified accounts.

**Table 2.** Top 20 users in our dataset.

| Sr. # | Account | Title | Total Tweets | Is Verified? | Followers |
|-------|---------|-------|--------------|--------------|-----------|
| 1 | @corona_news | اخبار كورونا فيروس | 11,952 | No | 15.6K |
| 2 | @aawsat_news | صحيفة الشرق الاوسط | 11,579 | Yes | 4.3M |
| 3 | @aljawazatksa | الجوازات السعودية | 10,043 | Yes | 1.6M |
| 4 | @newssnapnet | NewsSnap | 6837 | No | 3.1K |
| 5 | @menafnarabic | MENAFN.com Arabic | 6447 | No | 1.3K |
| 6 | @newsemaratyah | اخبار الاماراتUAE NEWS | 5954 | Yes | 185.5K |
| 7 | @aljoman_center | مركز الجُمان | 5669 | Yes | 20.6K |
| 8 | @misrtalateen | صحيفة مصر تلاتين | 5324 | No | 1K |
| 9 | @alahram | الأهرامAlAhram | 5238 | Yes | 5.6M |
| 10 | @alahramgate | بوابة الأهرام | 5213 | Yes | 158.8K |
| 11 | @rtarabic | RTARABIC | 5181 | Yes | 5.3M |
| 12 | @alainbrk | العين الإخبارية ـ عاجل | 4926 | Yes | 71.8K |
| 13 | @alroeya | صحيفة الرؤية | 4818 | Yes | 748.9K |
| 14 | @kuna_ar | كــــــــوناKUNA | 4417 | Yes | 993K |
| 15 | @libanhuit | Liban8 | 4192 | Yes | 19.2K |
| 16 | @alghadtv | قناة الغد | 4011 | Yes | 153.3K |
| 17 | @arabi21news | عربي21 | 4002 | Yes | 919.3K |
| 18 | @ch23news | Channel 23 | 3916 | No | 5.5K |
| 19 | @newselmostaqbal | المستقبل | 3778 | No | 486 |
| 20 | @emaratalyoum | الإمارات اليوم | 3741 | Yes | 2.2M |

### 3.2. Corpus Statistics

Figures 2 and 3 show the distribution of tweets in our dataset on a daily and monthly basis, respectively. It is shown that 21 March 2020 was the day with the most tweeting with more than 150,000 tweets, while March is the top in terms of the number of tweets with more than 1.44 million tweets.

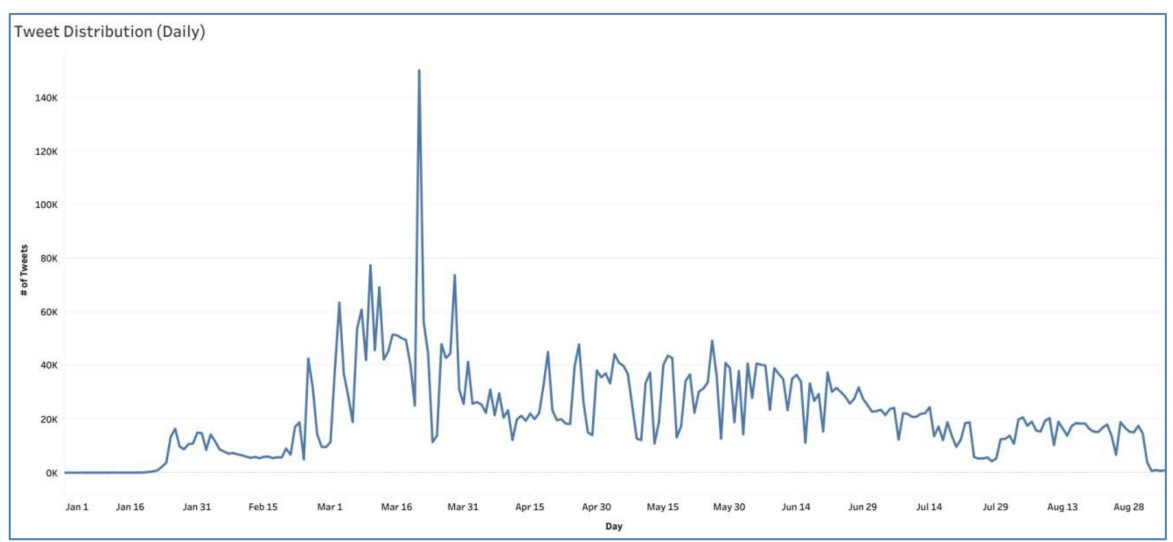

**Figure 2.** Tweets Distribution (Daily).

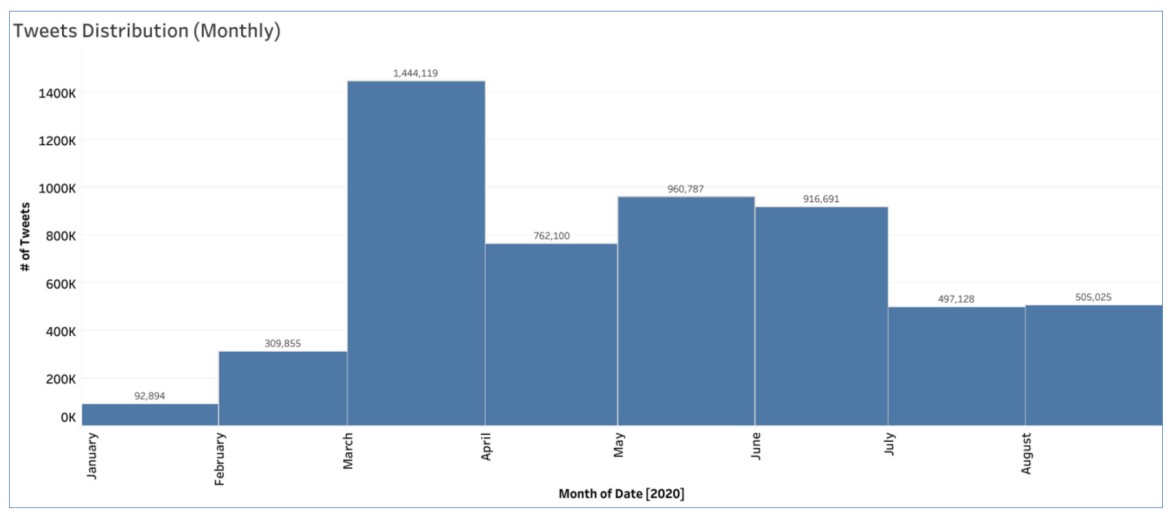

**Figure 3.** Tweets Distribution (Monthly).

The spike on 21 March is due to several reasons, such as the following:

- There was a total of 51,448 hashtags on that day.
- There was a total of 4819 unique hashtags on that day.
- By observing the top hashtags, we found that most of them were from Saudi Arabia where there was the imposition of closing shops and then issuing a curfew in large cities of the Kingdom.
- One day before that date, the Saudi government suspended all domestic flights, buses, taxis, and trains for 14 days.

### 3.3. Data Preprocessing

As the collected tweets were crawled from social media, the data are expected to be noisy and should be cleaned up before performing any of the NLP tasks to get better



results. The first step in text preprocessing is to remove URLs, hashtags, mentions, and media. Additionally, we performed a normalization process to unify the shape of some Arabic letters that have different shapes. Furthermore, the repeated characters problem was resolved by removing the extra repeated letters to return the word to its correct syntax. For example, the "كوروووونا" "Corona" is replaced with "كورونا" by removing the multiple occurrences of the character "و" with a single character. The diacritics are used to add decorations to the text especially with text posted on social media except for the text from the Holy Quran. These diacritics were removed from the tweets. Finally, the duplicated tweets were already excluded during the crawling process by looking for the tweet's ID.

### 3.4. Emotion Tweets Annotation

3.4.1. Rule-based Emotion Annotation

To annotate part of the dataset, we first used the six emotions of the Arabic emotion lexicon (https://github.com/motazsaad/emotion-lexiconfrom (accessed on 18 February 2021)) [37]. Detail about these lexicons is shown in Table 3. The lexicon words and phrases were used to retrieve tweets from the dataset using the rule-based (if-else) approach. We extracted 50,000 tweets against each emotion category. After performing emotion classification using the rule-based technique, we selected a sample of 300 tweets and annotated them manually to ensure the quality of the rule-based annotation technique. We also used an LSTM deep neural network to classify all tweets that were annotated using this annotation technique. The experiment and results are shown in the result section.

**Table 3.** Emotion lexicons statistics.

| Emotion | Arabic Words/Phrase |
|---------|---------------------|
| Anger | 748 |
| Disgust | 155 |
| Fear | 425 |
| Joy | 1156 |
| Sadness | 522 |
| Surprise | 201 |
| Total | 3207 |

3.4.2. Automatic Emotion Annotation

We used a FastText (https://fasttext.cc/ (accessed on 18 February 2021)), a neural network library to automatically annotate the unlabeled tweets (around 5.2 million) as illustrated in Figure 4. The FastText algorithm is an open-source NLP library developed by Facebook AI. It is a fast and excellent tool to build NLP models and generate live predictions. The algorithm was tuned with the following parameters: a learning rate of 0.5, an n-gram of 1, and the number of epochs of 50. We selected this algorithm because we are dealing with a large dataset where the prediction task using the traditional machine learning algorithms is very slow.

We fed the FastText algorithm with tweets that had been annotated using emotion lexicons from the previous section. A FastText model was built using the input tweets and the unlabeled tweets passed through the created model to predict their emotion classes and probabilities towards the predicted classes, as shown in Figure 4. To ensure the quality of the new annotation approach, we defined a threshold value that considers tweets that have a probability value greater than or equal to the threshold to be added to the new labeled tweets to expand the rule-based dataset. The tweets that have a probability value less than the defined threshold were ignored as these tweets have a lower confidence value and they are mostly neutral. The threshold value was selected carefully after performing several experiments as described in the experiment section. We also trained a deep learning classifier to test the quality of the expanded dataset. Algorithm 1 shows the pseudo code explaining how the automatic emotion annotation worked.

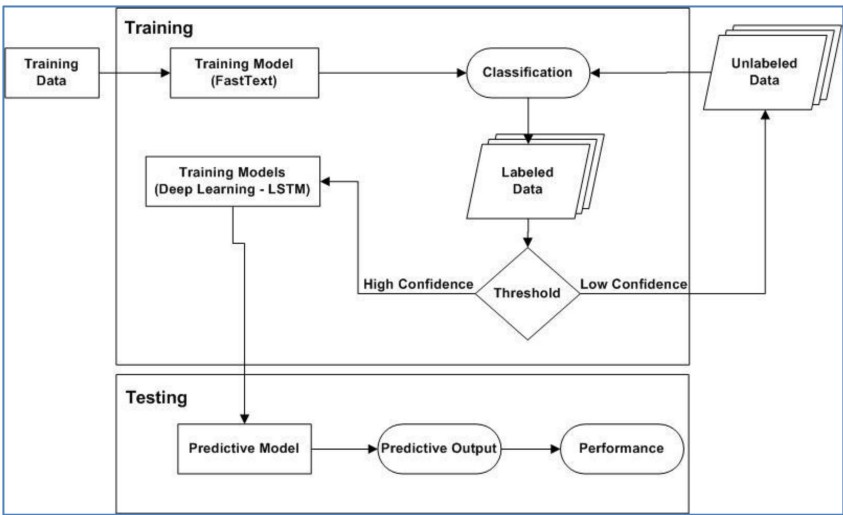

**Figure 4.** Automatic Emotion Annotation.

---

**Algorithm 1** Automatic Emotion Annotation Algorithm

---

**Data:** (LabeledData (300K Tweets), UnlabeledData (5.2M Tweets))
**Result:** NewLabeledData
TrainingSet = LabeledData;
UnlabeledData = UnlabeledData;
NewLabeledData;
ThresholdValue = 0.8
//training fastText model on TrainingSet
fastTextModel = TrainClassifier (TrainingSet);
**while** *(t ≤ UnlabeledData.size()) * **do**
    *//predict most likely emotion classes of t from fastTextModel*
    *emotionClass = fastTextModel.predict(UnlabeledData(t));*
    *//predict most likely emotion probabilities of t from fastTextModel*
    *emotionClassProbability = fastTextModel.predict-prob(UnlabeledData(t));*
    **if***(emotionClassProbability ≥ ThresholdV alue)* **then**
     *NewLabeledData.Add(t and emotionClass);*
    **End**
**end**

---

*3.5. Symptom Tweets Annotation*

To detect tweets that had symptom keywords, we prepared a list of words that represent the COVID-19 symptoms manually, after reading and translating the symptoms keywords from the WHO website. For example, the symptom word "fever" "حمى" was considered as a keyword to extract all tweets mentioning this word along with its derivations such as الحمى،بالحمى،والحمى. We collected more than 500 keywords to be used as a lexicon for the COVID-19 symptoms. We then used a rule-based approach to classify the tweets into symptom or non-symptom using a set of "if-then" rules. A tweet was considered to be a symptom tweet if it contained one or more keywords. The following is a list of COVID-19 symptoms as mentioned on the WHO website:

Most common symptoms:

- Fever
- Tiredness
- Dry cough

Less common symptoms:

- Loss of smell or taste
- Pains and aches

- Headache
- Sore throat
- Diarrhea
- Conjunctivitis
- Rash on skin
- Discoloration of fingers or toes

  Serious symptoms:

- Difficulty breathing or shortness of breath
- Chest pain or pressure
- Loss of speech or movement

### 3.6. Deep Learning Architecture

Deep neural networks are used to classify images, speech, and text using multiple processing layers with non-linear transformations. They can model high-level abstraction in data. LSTM is a special type of recurrent neural network (RNN) that is suitable for solving problems that require sequential information such as text classification tasks, using units with internal states that can remember information for long periods.

The neural network used in this research employs word embeddings with LSTM to perform the contextual text classification tasks. The preprocessed tweets were fed to the neural network, and we used the emotion classes to perform supervised learning on the tweets. Figure 5 shows the neural network layers and we describe each layer below.

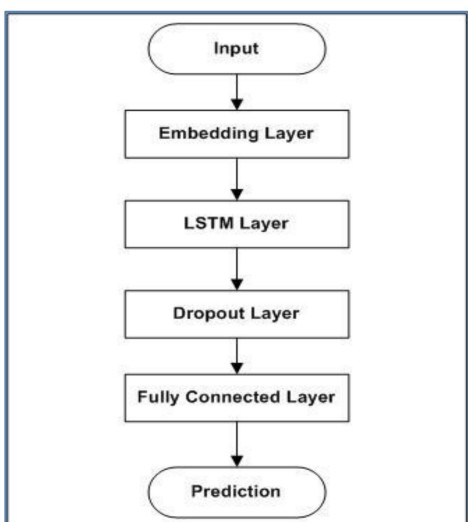

**Figure 5.** Deep Learning Architecture.

### 3.6.1. Embedding Layer

The first layer in the network is the embedding layer that converts integer indices in the input into a dense real valued vector of fixed size. The embedding layer learns a mapping that embeds each word in the discrete vocabulary to a lower dimensional continuous vector space. The use of this layer enables the extraction of semantic features from the input without performing a manual definition of features. The output of the embedding layer is fed to the next layer in the network.

### 3.6.2. LSTM Layer

LSTM layer is the second layer in the network. The LSTM unit is a memory cell composed of four main components: 1) input gate, 2) self-recurrent connection, 3) forget gate, and 4) output gate. The input gate decides what new information to store in the cell state. The forget gate allows the cell state to remember or forget its previous state by

controlling the cell's self-recurrent connection. Similar to the input gate, the output gate either allows or prevents the cell state from affecting other units.

### 3.6.3. Dropout Layer

To avoid overfitting or dropout, a regularization technique is used by randomly dropping a fraction of the units while in the training step. This layer prevents units from co-adapting.

### 3.6.4. Fully Connected Layer

This layer is fully connected to all of the activations in the dropout layer. It is used to learn non-linear combinations of the high-level features learned by the previous layers.

### 3.6.5. Loss Layer

The last layer in the network is the loss layer. This layer determines how the deviations of the predicted classes from the actual classes are penalized. Since we are interested in both the multi-class and binary classification of tweets, we use *binary_crossentropy* and *categorical_crossentropy*as the loss functions in emotion and symptom classification, respectively.

## 4. Experiments

### 4.1. Dataset

We used a dataset consisting of 5.5 million tweets to perform both emotion and symptom classification as described in the "data collection" section. Table 4 shows statistical details about the dataset. The dataset contains more than 100 million words and 2.65 million unique words.

**Table 4.** Corpus Statistics.

| Title | Number |
|---|---|
| Total Tweets | 5,499,318 |
| Total Words | 100,788,175 |
| Unique Words | 2,657,173 |
| Unique Users | 1,402,874 |
| Average Words per Tweet | 18.3 |

### 4.2. Experimental Setup

In this research, we performed multiclass classification (emotion detection) and binary classification (symptom and non-symptom detection). For the training and classification, an LSTM deep learning classifier was used to perform our experiments. We mapped each tweet in the corpus into a word embedding, which is a popular technique when working with text with a sequence length of 300 and embedding dimension of 100. We also limited the total number of words that we were interested in modeling to the 20,000 most frequent words. As described before, LSTM is a special kind of recurrent artificial neural network (RNN) architecture used in deep learning. It was designed to avoid the long-term dependency problem by remembering information for long periods. This makes LSTM suitable for problems that require sequential information, such as text processing tasks.

### 4.3. Evaluation Metrices

All the results are reported using the accuracy, precision, recall, and F1-measureas follows:

$$\text{Accuracy} = \frac{\text{TP} + \text{TN}}{(\text{TP} + \text{TN} + \text{FP} + \text{FN})} \tag{1}$$

$$\text{Precision} = \frac{\text{TP}}{(\text{TP} + \text{FP})} \tag{2}$$

$$Recall = \frac{TP}{(TP + FN)} \tag{3}$$

$$F1 - measure = 2 \times \frac{(Precision \times Recall)}{(Precision + Recall)} \tag{4}$$

where the TP is the cases in which we predicted YES and the actual output was also YES, FP is the cases in which we predicted YES and the actual output was NO, TN is the cases in which we predicted NO and the actual output was NO, and FN is the cases in which we predicted NO and the actual output was YES.

### 4.4. Emotion Classification Results

#### 4.4.1. Rule-Based Classification Results

In this section, the output of the rule-based classification, containing 300,000 tweets and representing all emotion categories (six emotion categories and 50,000 tweets for each category), was used to perform the first experiment. The sample tweets represent 5.46% of the dataset. The tweets were used to train an LSTM deep learning model (80% training and 20% testing). The results of the rule-based emotion classification are shown in Figure 6.

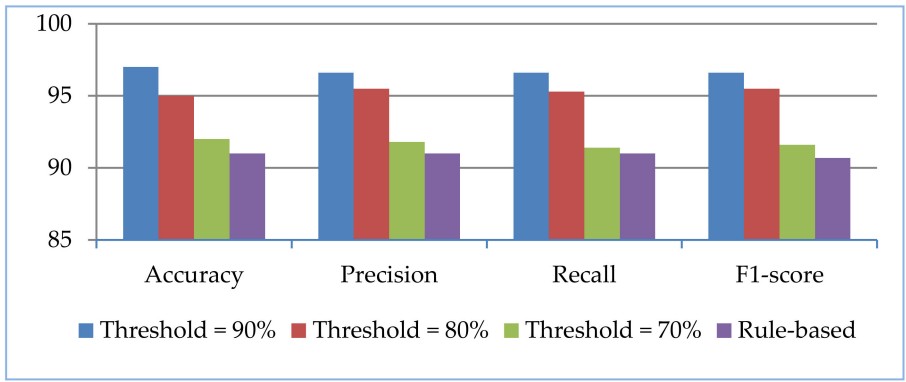

**Figure 6.** LSTM Classification Results.

To evaluate the results obtained from the rule-based annotation, we randomly selected a list contains 360 tweets (60 tweets in each emotion class) from the large dataset. Two human annotators were asked to annotate the given tweets. Lists of guidelines were given to the annotators to understand how to annotate the sample list correctly. Both annotators agreed to annotate 300 tweets with the same emotion classes. We compared the results obtained from the rule-based annotation with the ground truth sample tweets which were annotated by humans. The confusion matrix of the comparison between the ground truth and the results obtained from the automatic annotation is shown in Table 5.

**Table 5.** Confusion matrix of the comparison between ground truth and automatic annotation.

|  | Anger | Disgust | Fear | Joy | Sadness | Surprise |
|---|---|---|---|---|---|---|
| **Anger** | 40 | 3 | 2 | 1 | 1 | 3 |
| **Disgust** | 4 | 38 | 2 | 2 | 3 | 1 |
| **Fear** | 1 | 4 | 41 | 3 | 1 | 0 |
| **Joy** | 2 | 2 | 0 | 44 | 1 | 1 |
| **Sadness** | 2 | 1 | 1 | 3 | 42 | 1 |
| **Surprise** | 1 | 1 | 1 | 3 | 1 | 43 |

In Table 6, the results show that our rule-based emotion classification achieved~83% F1-score by comparing the results of rule-based annotation with the human annotation using 300 tweets.

**Table 6.** Results evaluation.

| # | Class | Precision | Recall | F1 Score |
|---|-------|-----------|--------|----------|
| 1 | Anger | 0.80 | 0.80 | 0.80 |
| 2 | Disgust | 0.76 | 0.78 | 0.77 |
| 3 | Fear | 0.82 | 0.87 | 0.85 |
| 4 | Joy | 0.88 | 0.79 | 0.83 |
| 5 | Sadness | 0.84 | 0.86 | 0.85 |
| 6 | Surprise | 0.86 | 0.88 | 0.87 |
| | Average | 0.826 | 0.83 | 0.828 |

### 4.4.2. Automatic Classification Results

The automatic annotation was explained in the annotation section. The tweets that have a confidence value greater than or equal to the defined threshold were added to represent the automatically annotated dataset. We perform this step to avoid neutral tweets and consider only tweets that have a high probability toward any of the emotion categories. This step helps in to ensure the quality of the developed corpus. We used the LSTM classifier to perform several experiments to obtain the best threshold value. Table 7 shows the distribution of emotion tweets after applying the threshold condition on the probability of emotion classes towards the tweets in our dataset. From the tweets distribution table below, it is clearly shown that increasing the threshold value decreases the number of tweets in the dataset and vice versa. Applying 90, 80, and 70% threshold values lead to 1.2 million, 1.67 million, and 2.16 million tweets, respectively. Each dataset was split into 80 and 20% for training and testing, respectively. Three classification experiments were conducted on these three datasets. Figure 6 shows that the classification results improved after expanding the rule-based emotion dataset. It shows that increasing the threshold value leads to increasing the quality of the classification results.

**Table 7.** Emotion tweets distribution.

| Sr. # | Emotion | Threshold (90%) | Threshold (80%) | Threshold (70%) |
|-------|---------|-----------------|-----------------|-----------------|
| 1 | Anger | 213,189 | 297,781 | 381,629 |
| 2 | Disgust | 206,025 | 330,530 | 417,140 |
| 3 | Fear | 231,869 | 326,931 | 421,748 |
| 4 | Joy | 241,264 | 283,606 | 406,080 |
| 5 | Sadness | 197,406 | 280,685 | 358,142 |
| 6 | Surprise | 119,198 | 151,022 | 185,202 |
| | Total Tweets | 1,208,951 | 1,670,555 | 2,169,941 |

The below figure shows that selection of a 90% threshold value gives higher results and fewer tweets. Similarly, decreasing the threshold value gives low results and more tweets. The results show that we can improve the results from 90% (F1-score) using the rule-based technique to 97% (F1-score) using the automatic annotation technique.

### 4.5. Symptom Classification Results

For the symptom classification, we first use a total of 200,000 tweets to train CNN and LSTM models. The sample tweets represent 3.64% of the dataset. The tweets were labeled using a dictionary of symptom words using a rule-based approach as described in the methodology. The result of symptom classification using an LSTM deep learning classifier after splitting the dataset into 80% for training and 20% for testing is shown in Table 8.

**Table 8.** Symptom Classification Results.

| Deep Learning Classifier | Accuracy | F1-Score (Macro Avg) | F1-Score (Weighted Avg) |
|---|---|---|---|
| LSTM | 0.75 | 0.75 | 0.75 |

Similarly, as in automatic emotion classification, we use a neural network to classify unlabeled tweets in our dataset automatically. Table 9 shows the distribution and percentage of symptom and non-symptom tweets. It is shown that the symptom tweets represent almost one-third of the dataset with more than 2 million tweets.

**Table 9.** Symptom and non-symptom tweets distribution.

| # | Type | Number of Tweets | % |
|---|---|---|---|
| 1 | Symptom Tweets | 2,034,748 | 37% |
| 2 | Non-symptom Tweets | 3,464,570 | 63% |
| | Total Tweets | 5,499,318 | |

*4.6. Monitoring System*

Emotion monitoring from social media data helps in tracking the distribution of emotions, symptoms, and user interactions.

4.6.1. Monitoring Emotion Distribution

In this section, we show the distribution of tweets in six emotion categories as depicted in Figure 7. It is shown that the day of March 21 had the highest number of tweets in all emotion categories (Anger = 9571, Disgust = 5311, Fear = 8279, Joy = 23,098, Sadness = 5869, Surprise = 4120). It is also clear that public emotions decreased over time. People's feelings are variable, but were at their highest during March.

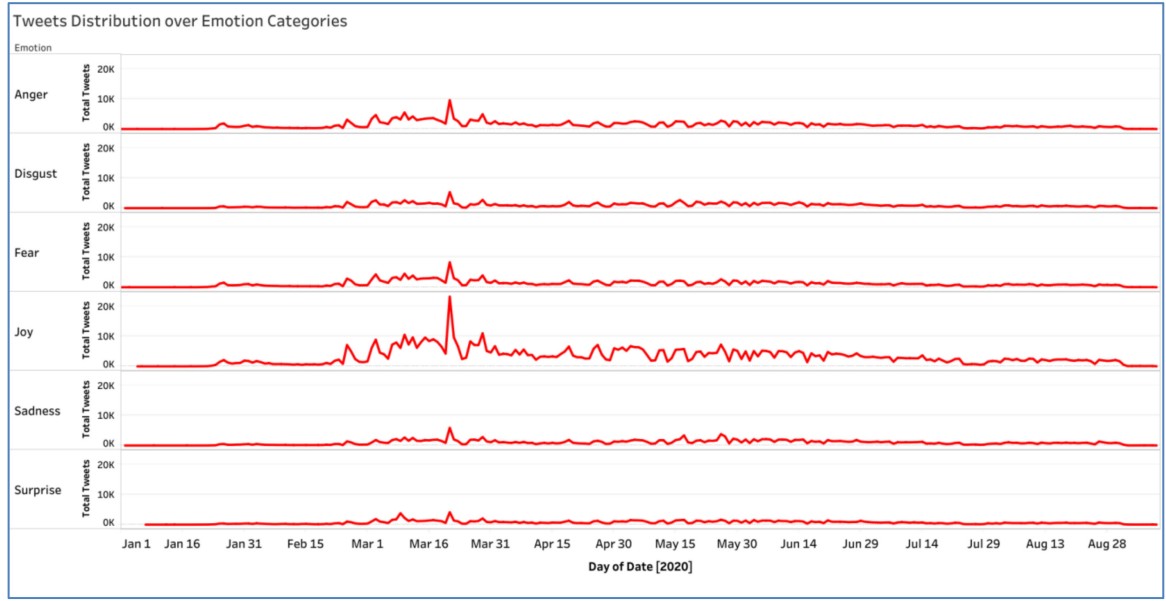

**Figure 7.** Emotion Distribution.

### 4.6.2. Monitoring Symptom Distribution

The symptom and non-symptom tweet distributions are shown in Table 9. Symptom and non-symptom tweets distribution. The 21st of March has the highest number of tweets that mention symptom keywords with 56,242 tweets, as shown in Figure 8.

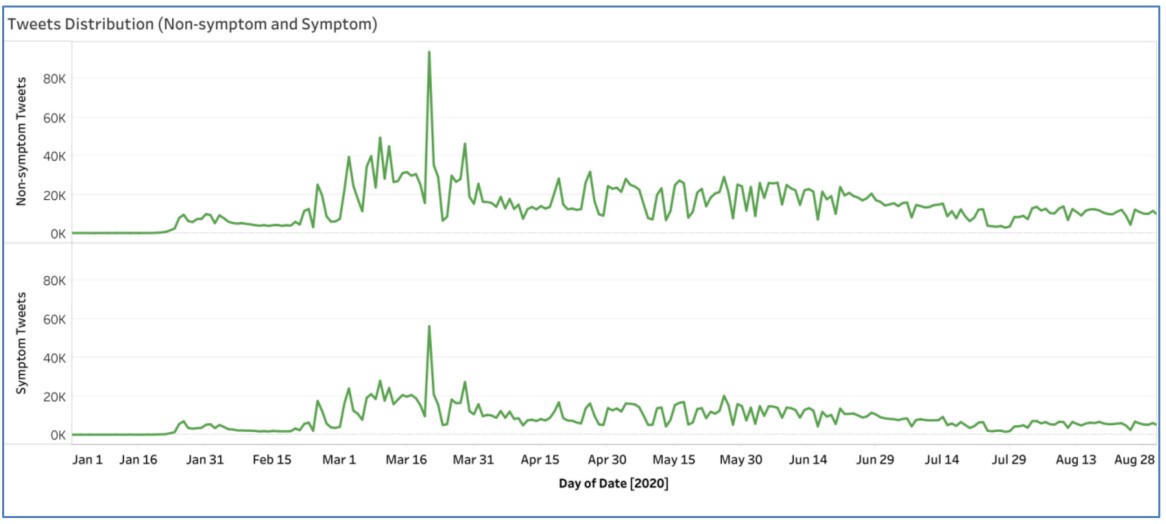

**Figure 8.** Symptom Distribution.

### 4.6.3. Monitoring User Interactions

In this section, we illustrate the tracking of the user interaction with all tweets in our dataset and the user interaction with emotions and symptoms tweets. In Figure 9, most of the user interactions are likes and re-tweets with 49.2 million and 15.4 million, respectively. The total number of replies is 5.8 million. The highest number of interactions (likes, re-tweets, and replies) was on 21 March with 1.46 million, 500,000, and 200,000, respectively.

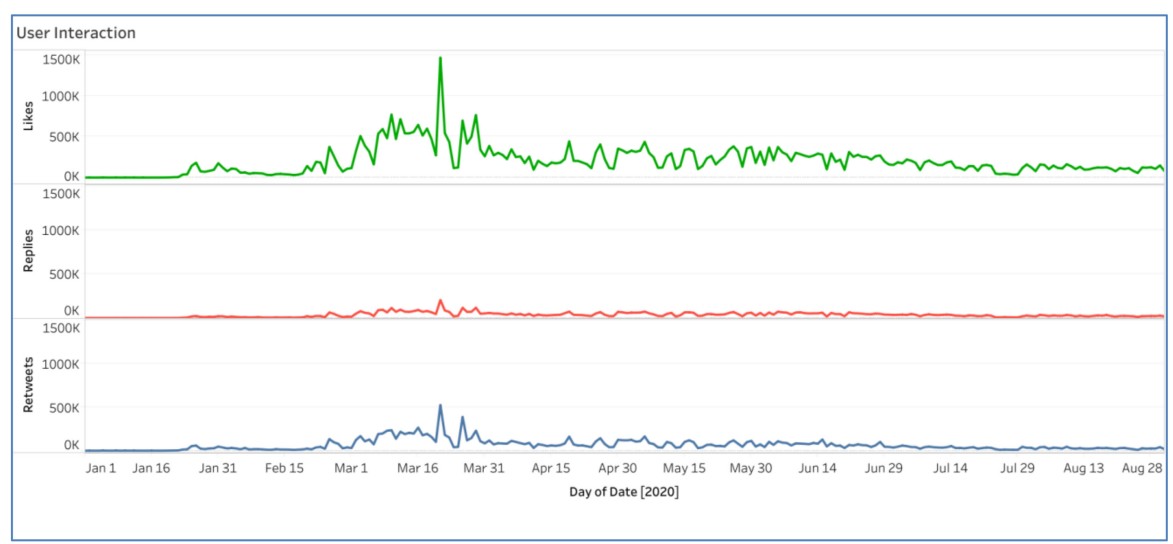

**Figure 9.** User Interaction.

In Figure 10, we show the user interactions with emotion categories. The joy category constitutes 36% of the total tweets in our dataset; it is the highest in terms of the number of likes, replies, and re-tweets with 22 million, 4 million, and 7 million, respectively. The second largest category in terms of the number of likes, replies, and re-tweets is the anger category with 8 million, 1.2 million, and 2.4 million, respectively. The third largest category

is the fear category with 6.4 million, 1 million, and 1.9 million likes, replies, and re-tweets, respectively. The categories disgust, sadness, and surprise came in fourth, fifth, and sixth, respectively.

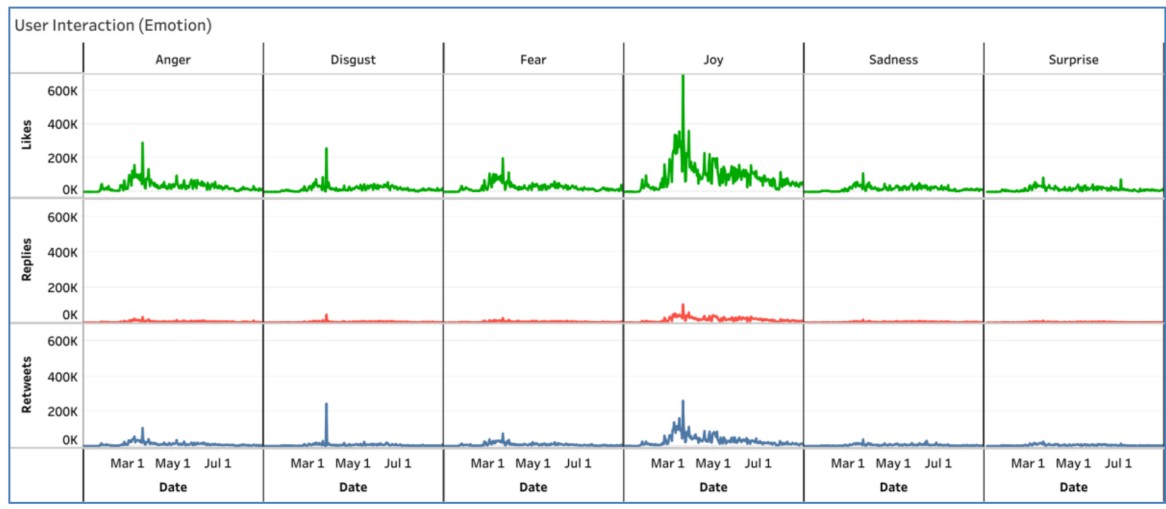

**Figure 10.** User Interaction (Emotions).

In Figure 11, we show the user interaction with non-symptom and symptom tweets. One of the findings is that the users interact with non-symptom tweets using likes and replies while they interact with symptom tweets using re-tweet, which means that users propagate most of the tweets that mention any of the COVID-19 symptoms to warn others. They interact with non-symptom tweets with 25.9 million, 4.4 million, and 7.1 million while they interact with symptom tweets with 23.4 million, 4.1 million, and 8.3 million likes, replies, and re-tweets, respectively.

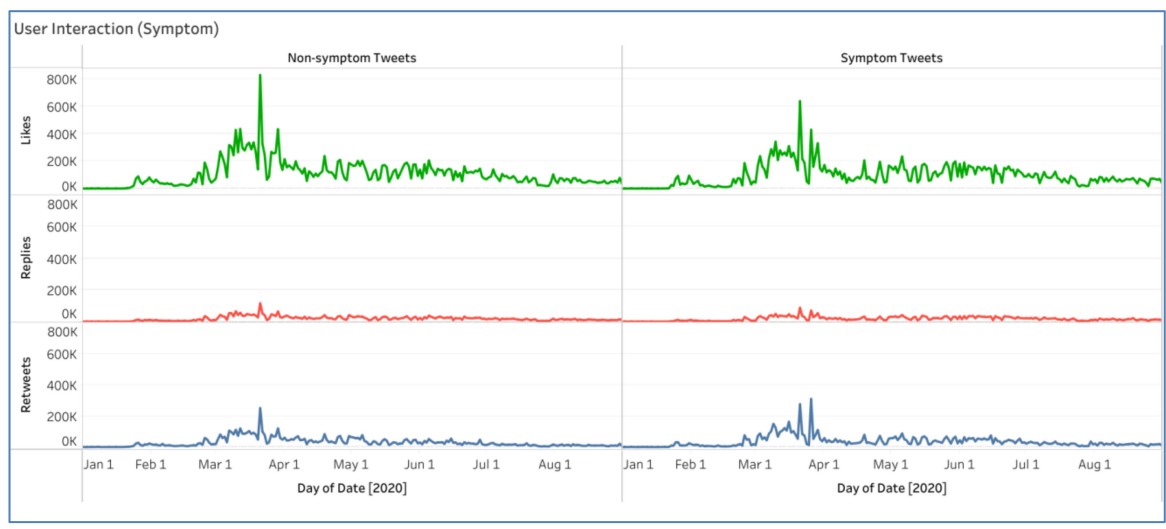

**Figure 11.** User Interaction (Symptoms).

## 5. Use Cases

As we are concerned with few emotion categories such as fear and anger along with tweets that mentioned COVID-19 symptoms, we will answer the following questions:

1. What do people fear?
2. Why are people angry?
3. What are the symptoms that cause people anxiety and fear?

To answer the above questions, we need to extract the discussed topics during the pandemic in both the fear and anger emotion categories and the topics of symptom tweets as well to understand people's concerns during the pandemic.

### 5.1. Anger Emotion Tweets

As the anger emotion came second in terms of the number of tweets and user interactions, we will show the most discussed topics in this category in order to understand why people are angry. We selected the tweets in March as there are 1.44 million tweets. We used Latent Dirichlet Allocation (LDA), a topic modeling technique to extract the topics discussed during March. We used perplexity to estimate the optimal number of topics as shown in Figure 12. The graph shows that with eight topics, we got the highest coherence score of 0.364.

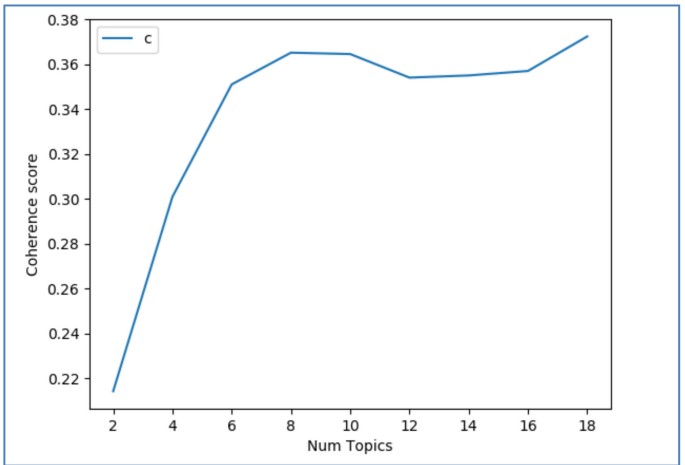

**Figure 12.** The optimal number of topics (Anger Tweets).

The following are the top eight most discussed topics:
- An increase in COVID-19-infected people due to the outbreak of the corona virus epidemic.
- The call to stay at home to curb the spread of the epidemic.
- Wars continuing despite the spread of the epidemic in some countries, such as Yemen.
- Anger and accusations of China spreading the virus.
- The possibility of death from infection with the virus and neglect of governments.
- The carelessness of people during the time of the pandemic.
- Anger over China's deliberate transmission of the pandemic to Muslims.
- The risk of transmitting the pandemic via arrivals from Iran.

### 5.2. Fear Emotion Tweets

The fear category came third in terms of the number of tweets and user interactions. Similar to the detection of anger topics, we use LDA to detect the discussed topics during March. We used perplexity to estimate the optimal number of topics, as shown in Figure 13. The graph shows that the highest coherence value was obtained with four topics.

The following are the top five topics that are discussed as follows:
- The government's role in fighting the epidemic.
- The collapse of countries' economies due to the closure of borders.
- The fear of infection and death from the virus and praying to God to raise the epidemic.
- Fear of the long stay-at-home quarantine.

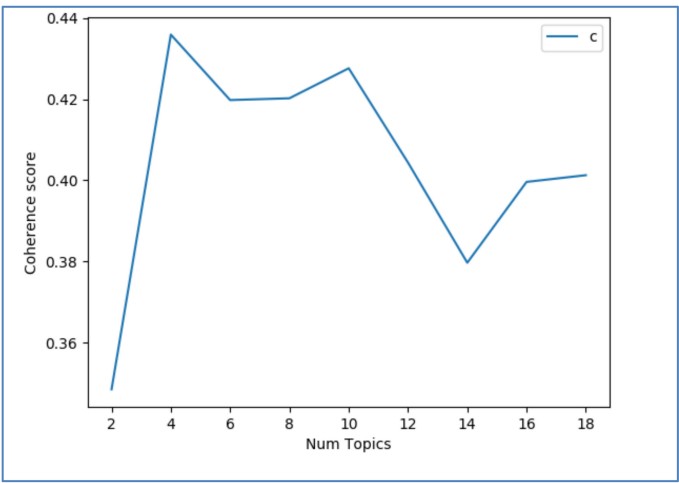

**Figure 13.** The optimal number of topics (fear tweets).

### 5.3. Symptom Tweets

We used LDA to detect the top five topics discussed in March. We used perplexity to estimate the optimal number of topics, as shown in Figure 14. The graph shown is that the optimal number of topics is four. We got a coherence score of "0.415".

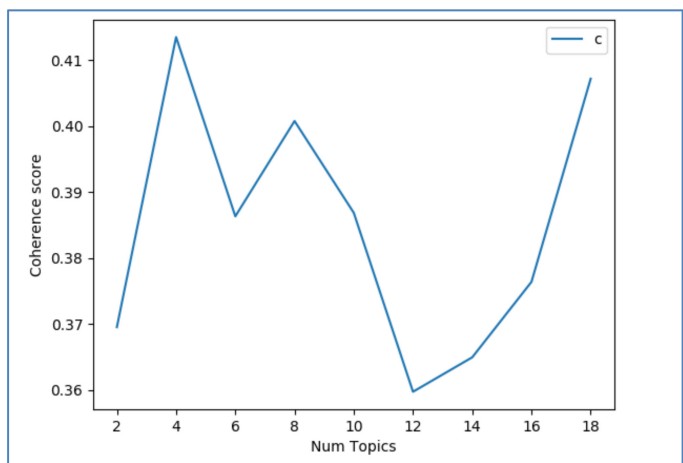

**Figure 14.** The optimal number of topics (Symptom Tweets).

The following are the detected topics:

- Discussion about disease and treatment.
- Discussion about ways to prevent corona.
- Prayers for healing for the COVID-19-infected people.
- Exposure to some symptoms of infection with the corona virus, such as the throat.

### 6. Discussion and Limitation

The major finding in our research is to develop the first Arabic emotion dataset for monitoring people's mental health issues during theCOVID-19 pandemic. We address these using COVID-19 messages on Twitter. In this research, we are concerned with the analysis of six emotion categories only between 1 January and 31 August 2020.

A combined approach of rule-based (Lexicon-based) and automatic annotation (neural networks) was useful for annotating our dataset. The results showed several essential points. First, the rule-based annotation approach helps in the annotation of part of our dataset and the outcomes of this step are a total of 300,000 tweets distributed among six emotion classes. The manual verification of the rule-based annotation in this step gives

an 83% F1-score, which is promising compared with the size of the dataset. Second, the automatic annotation using neural networks is used to expand the dataset, which was annotated using the rule-based approach. The results of expansion show better results if we increase the probability of an emotion class towards the tweet. It also shows that we can improve the emotion classification from rule-based by 5.9, 4.8, and 0.9% using threshold values of 90, 80, and 70%, respectively.

The monitoring system shows that most of the tweets in our dataset were posted in March in both emotion and symptom tweets. It is also shown that more tweets are reflect people's anger and fear. The system also depicted that people interact with COVID-19 non-symptom tweets using likes and replies while they use re-tweets to share the tweets that mention COVID-19 symptoms. The study concludes with multiple points that explain the cause of anger and fear during the pandemic.

### 7. Conclusion and Future Work

In this research, we present a monitoring system for people's emotions and symptoms mentions during the COVID-19 pandemic. We use Twitter as a data source to collect 5.5 million tweets spanning from January to August 2020. Each tweet in the dataset was labeled with the emotion category—anger, disgust, fear, joy, sadness, and surprise— and symptom or non-symptom. We used rule-based and neural network approaches to annotate our dataset using emotion and symptom lexicons. We used these annotated tweets to build multiple deep learning models using an LSTM neural network to verify the quality of the datasets. After annotating all of the tweets in our dataset, we built a monitoring system to track the people's emotions and symptom mentions to understand the change in people's emotions and extract more symptoms.

Our findings facilitate the understanding of people's emotions during the epidemic and can be used to track people's emotions which lead to or indicate mental health issues. The monitoring system should help governments, health authorities, and decision-makers to reassure people and dispel their fear of the pandemic, in cooperation with psychologists, psychiatrists, and doctors.

In the future, and as our emotion categorization is limited to six emotion classes, we are planning to expand the emotion classes by adding more emotion categories. We also plan to build a web-based monitoring system that will crawl tweets and annotate tweets in real-time. The system will be helpful in monitoring people's emotions during any future epidemic.

**Author Contributions:** Conceptualization, A.A.-L.; investigation, M.A.; methodology, A.A.-L.; project administration, M.A.; resources, A.A.-L.; software, A.A.-L.; supervision, M.A.; validation, M.A.; visualization, A.A.-L.; writing—original draft, A.A.-L.; writing—review & editing, M.A. All authors have read and agreed to the published version of the manuscript.

**Funding:** This work was supported by Prince Sultan University, Riyadh, Saudi Arabia.

**Institutional Review Board Statement:** Not applicable.

**Informed Consent Statement:** Not applicable.

**Data Availability Statement:** The code datasets are freely available for research purposes in this link: https://github.com/yemen2016/COVID-19-Arabic-Emotion-Dataset (accessed on 18 February 2021).

**Acknowledgments:** Authors are thankful to Prince Sultan University, Saudi Arabia for providing the fund to carry out the work.

**Conflicts of Interest:** The authors declare that they have no conflicts of interest to report regarding the present study.

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
