# Peer review of "Monitoring People’s Emotions and Symptoms from Arabic Tweets during the COVID-19 Pandemic"

_information, doi:10.3390/info12020086_

Round 1
Reviewer 1 Report
The major contributions in the conducted work are the development of a system for monitoring people's emotions (based on the Arabic content) and the use of this to harvest a large Covid-19 Twitter corpus and automatically classify it (using deep learning-based algorithms) to allow for some analysis. I find the focus on Arabic to be interesting (as most related research is conducted on tweets written in English), but there are methodological concerns that make me question the validity of the analysis results.
In Section 2, the authors aim to present a literature survey of "the existing monitoring systems". Much interesting work is discussed, but it should be noted that it is far from exhaustive. The literature survey is not argued to be the most important contribution, but make clear to the reader that it is not exhaustive, or that it is targeted only on systems being used to analyse Covid-19 related tweets. Examples of relevant journal articles describing this kind of systems outside the Covid-19 context can for example be found in "Security Informatics", but there are many other relevant systems as well which are not covered in the survey.
My main concern is related to the methodology (Section 3). The automatic classification of the emotion of tweets in Arabic is at the core of the work, but this is not solid enough. On a high level it is described as follows: "Arabic emotion lexicons were used to manually annotate a list of 300K tweets. We then use deep learning classifiers to ensure the quality of the manually annotated tweets. The best deep learning model was used to annotate the large unlabeled dataset." Both the "manual annotation" and the developed deep learning classifiers have severe problems. First of all, the manual annotation does not seem to be manual, but rather relying fully on a Arabic emotion lexicon. The quality of this wordlis/ rule-based annotation (or the resulting deep learning classifiers) are never tested against a ground truth (such as tweets being annotated using a team of human annotators). Therefore, we only can tell how good the classifiers are at predicting the label given by an emotion lexicon-based approach, rather than the true emotion expressed in tweets. Moreover, there are way too few details presented regarding the classifiers. To even use a CNN for text seems as a very unorthodox choice. LSTMs are more reasonable, but what about a Transfomer-architecture such as BERT? I also lack details such as if you represent the tweets using word embeddings, one-hot encoding, etc. as this can have a huge effect on the results.
You mention that you plan to annotate sample tweets by the human to get more accurate results. In order for the reader to be able to get an idea of how good the classifier works (and thereby to which degree the analysis results can be trusted), I would argue that this cannot be left for future work, but rather has to be done before this work can be published anywhere.
Other issues that can be seen in many studies on social media data are the impact of the choice of keywords (to identify tweets to collect and analyze) and the effect this can have on the results, together with the bias of analyzing tweets (over- and underrepresentation of certain demographic subparts of the population). This is not specfic to this study, but would be good to reflect on more in the discussion of the obtained results. I am not Arabic-speaking, but there are potential privacy issues with presenting lists of accounts / users in this way.
Although I am not a native English speaker (and therefore do not want to review the language in great detail), there certainly is room for improvement of the language in the manuscript. A few examples of mistakes are the following:
- "specific regions and notional wide"
- "tweets that mention any of COVID-19 symptoms."
- "The rapidly spreading creating a global public health crisis"
- "we use to take advantage of the emotional reactions"
- "We used to classify the dataset into six different emo-
tions." - "In this research, we use to take advantage of the emotional reactions"
These are just a few examples from the first pages of the manuscript, but it should be enough to highlight that some more oversight of the language has to be done.
Author Response
Response to Reviewer 1 Comments
We would like to thank the reviewers for dedicating the time to go through our paper. Please find below our response to his valuable comments:
Point 1: In Section 2, the authors aim to present a literature survey of "the existing monitoring systems". Much interesting work is discussed, but it should be noted that it is far from exhaustive. The literature survey is not argued to be the most important contribution, but make clear to the reader that it is not exhaustive, or that it is targeted only on systems being used to analyse Covid-19 related tweets. Examples of relevant journal articles describing this kind of systems outside the Covid-19 context can for example be found in "Security Informatics", but there are many other relevant systems as well which are not covered in the survey.
Response 1: Yes. We added more details in the literature by providing more tracking systems in different domains including tracking news entities, spikes, disasters, tourists, and elections.
Point 2:My main concern is related to the methodology (Section 3). The automatic classification of the emotion of tweets in Arabic is at the core of the work, but this is not solid enough. On a high level it is described as follows: "Arabic emotion lexicons were used to manually annotate a list of 300K tweets. We then use deep learning classifiers to ensure the quality of the manually annotated tweets. The best deep learning model was used to annotate the large unlabeled dataset." Both the "manual annotation" and the developed deep learning classifiers have severe problems. First of all, the manual annotation does not seem to be manual, but rather relying fully on a Arabic emotion lexicon. The quality of this wordlis/ rule-based annotation (or the resulting deep learning classifiers) are never tested against a ground truth (such as tweets being annotated using a team of human annotators). Therefore, we only can tell how good the classifiers are at predicting the label given by an emotion lexicon-based approach, rather than the true emotion expressed in tweets. Moreover, there are way too few details presented regarding the classifiers. To even use a CNN for text seems as a very unorthodox choice. LSTMs are more reasonable, but what about a Transfomer-architecture such as BERT? I also lack details such as if you represent the tweets using word embeddings, one-hot encoding, etc. as this can have a huge effect on the results.
Response 2:
It was a mistake of writing manual annotation which is, in fact, a rule-based approach. We update the paper with rule-based instead of manual annotation.
We have selected a sample of 300 tweets randomly from the automatically annotated corpus and manually annotate them using two human annotators. We used the manual annotated tweets as ground truth to check the quality of our automatic annotated corpus.
We also elaborated the classification details.
The input of both deep learnings is word embedding. A detail about this is added in the experimental setup section.
Point 3:You mention that you plan to annotate sample tweets by the human to get more accurate results. In order for the reader to be able to get an idea of how good the classifier works (and thereby to which degree the analysis results can be trusted), I would argue that this cannot be left for future work, but rather has to be done before this work can be published anywhere.
Response 3:Yes. We employed two annotators to manually annotate a sample of 300 tweets to be used as ground truth to test the quality of our proposed annotation techniques.
Point4:Other issues that can be seen in many studies on social media data are the impact of the choice of keywords (to identify tweets to collect and analyze) and the effect this can have on the results, together with the bias of analyzing tweets (over- and underrepresentation of certain demographic subparts of the population). This is not specific to this study, but would be good to reflect on more in the discussion of the obtained results. I am not Arabic-speaking, but there are potential privacy issues with presenting lists of accounts / users in this way.
Response4:we selected the keywords by observing the top hashtags in most of the Arab countries. In our research and as it is in T&C of Twitter, we are not allowed to share the user’s information. The only thing which we can share is the tweet’s ids only.
Point5:Although I am not a native English speaker (and therefore do not want to review the language in great detail), there certainly is room for improvement of the language in the manuscript. A few examples of mistakes are the following:
"specific regions and notional wide"
"tweets that mention any of COVID-19 symptoms."
"The rapidly spreading creating a global public health crisis"
"we use to take advantage of the emotional reactions"
"We used to classify the dataset into six different emotions."
"In this research, we use to take advantage of the emotional reactions"
These are just a few examples from the first pages of the manuscript, but it should be enough to highlight that some more oversight of the language has to be done.
Response5: The manuscript is revised by the English editing services in our university to resolve the English mistakes.
Reviewer 2 Report
Dear authors
The article sounds interesting for the readers. You made a great effort to prepare and present your study. There are some concerns about the article that you can work on them for the probable revision. There is a political concern for me about your main sources. For example, you are following texts about corona in Iran as well. Iranians are not Arabian. I can understand some biases on Iran in the many Arabic countries, but why not other countries? Some Arabic countries have more infections based on their population in comparison to Iran and many other countries. The discussion and conclusion parts are weak. We have information right now, and we need to know what we can do with them. I expect a really better discussion and conclusion parts. There are many errors in the reference list, mostly about writing the full names of all authors. Please describe clearly all targeted countries of your study. The time period needs to be described clearer.
Author Response
Response to Reviewer 2 Comments
Point 1: The article sounds interesting for the readers. You made a great effort to prepare and present your study. There are some concerns about the article that you can work on them for the probable revision. There is a political concern for me about your main sources. For example, you are following texts about corona in Iran as well. Iranians are not Arabian. I can understand some biases on Iran in the many Arabic countries, but why not other countries? Some Arabic countries have more infections based on their population in comparison to Iran and many other countries. The discussion and conclusion parts are weak. We have information right now, and we need to know what we can do with them. I expect a really better discussion and conclusion parts. There are many errors in the reference list, mostly about writing the full names of all authors. Please describe clearly all targeted countries of your study. The time period needs to be described clearer.
Response 1:
We would like to thank the reviewers for dedicating the time to go through our paper. Please find below our response to his valuable comments:
- Our research aims to track people's emotions and provide the decision-makers with clear insights into people's emotions in order to understand people's fear or anger emotions. As many people from Arab regions travel to Iran, People have fears that the virus could be transmitted through people returning from this country. This is the reason for considering the tweets mentioning “Iran” keywords.
- We have improved the discussion and conclusion section.
- The issues on the references are resolved.
- References updated. We used endnote to arrange the references as suggested in the journal’s template.
- Conclusion and discussion parts updated.
Round 2
Reviewer 1 Report
Although the authors have made some attempts to strengthen the work, most concerns from previous reviews are still relevant.
Some manual annotation has been done to try to verify that the rule-based annotations are reasonable. This is a step in the right direction, but the rule-based annotation only seems to be a quick way to annotate large amounts of textual data. Since the authors build deep learning-based classifiers, it certainly seems like the authors see those as the more interesting classifiers. These are still only evaluated against the rule-based annotations. To use these automatic annotations as the benchmark is not very interesting, at least not on their own. To (over)fit a deep learning classifier so that it learns to make the same overly simplified classifications as the rule-based classifier would probably not be hard, but such a classifier would not be more interesting just because it achieves an accuracy of 1.00 when compared to the output of the rule-based classifier. It is obvious that better evaluations are needed for the trained classifiers, otherwise you can use the rule-based classifier directly without no added value of more sophisticated approaches.
I am also surpised to see that the shown confusion matrix only shows the six emotion classes of interest. Do you see these as exhaustive? I would expect that a large proportion of all tweets lack emotional content, so that the ground-truth of these tweets should be neutral.
The added details of the experimental setup are still not very informative. Currently the experiments section and the vague descriptions of the used architectures etc. are badly described and make the reader question how these implementations and experiments have been made, I would encourage signficant improvement of the experiments section and, if possible, source code which allows other researchers to verify the soundness of your implementations and verifying the obtained results.
If you aim to have the manuscript published, a major revision is needed, not just simply adding a few sentences and changing a few words here and there.
Author Response
Point 1: Some manual annotation has been done to try to verify that the rule-based annotations are reasonable. This is a step in the right direction, but the rule-based annotation only seems to be a quick way to annotate large amounts of textual data. Since the authors build deep learning-based classifiers, it certainly seems like the authors see those as the more interesting classifiers. These are still only evaluated against the rule-based annotations. To use these automatic annotations as the benchmark is not very interesting, at least not on their own. To (over)fit a deep learning classifier so that it learns to make the same overly simplified classifications as the rule-based classifier would probably not be hard, but such a classifier would not be more interesting just because it achieves an accuracy of 1.00 when compared to the output of the rule-based classifier. It is obvious that better evaluations are needed for the trained classifiers, otherwise you can use the rule-based classifier directly without no added value of more sophisticated approaches.
Response1: We change our annotation approach by combining rule-based and neural network approaches to build our corpus. The former is used to annotate part of our dataset (300K tweets) with help of Arabic emotion lexicons. We verify the rule-based annotation by comparing subset of the annotation with human annotation as describe in the first round. To overcome the issue of neural tweets or tweets from mixed emotions we use neural network classifier to expand the rule-based annotated tweets by labeling each tweet from the unlabeled tweets with emotion class and probability of that emotion towards the tweet (e.g., tweet1 : Angry , 0.95). We only considered those tweets that have a probability value greater or equal a certain threshold after performing several experiments to select the best threshold value. The results of classification using the expanded dataset was improved using the same classifier as described in the experiment section.
Point 2: I am also surprised to see that the shown confusion matrix only shows the six emotion classes of interest. Do you see these as exhaustive? I would expect that a large proportion of all tweets lack emotional content, so that the ground-truth of these tweets should be neutral.
Response 2: In the ground truth we also eliminated of 60 tweets that are neutrals as described in the rule-based classification results section. We eliminated of more neutral tweets on the automatic annotation using neural network annotation approach by considering tweets that have probabilities greater than or equal to certain threshold as described in the automatic classification results.
Eliminated
of
Point 3: The added details of the experimental setup are still not very informative. Currently the experiments section and the vague descriptions of the used architectures etc. are badly described and make the reader question how these implementations and experiments have been made, I would encourage significant improvement of the experiments section and, if possible, source code which allows other researchers to verify the soundness of your implementations and verifying the obtained results.
Response 3: The detail about our experiment is further explained. We added a separate section explaining the deep neural network architecture which is used to perform our classification.
Source code and datasets are freely available online for research purposes in this link:
https://github.com/yemen2016/COVID‐19‐Arabic‐Emotion‐Dataset
Point 4: If you aim to have the manuscript published, a major revision is needed, not just simply adding a few sentences and changing a few words here and there.
Response 4: We really interested in publishing our work in your great journal. The updated manuscript is completely changed and we hope that we handle all of the issues.
Reviewer 2 Report
Dear authors
You didn't answer my main concerns. Also still, there are some errors in the reference list.
Author Response
Point 1: The previous comment in the first round:
Response 1: The collected text that mentioning "Iran" is written in Arabic and mostly by Arabic people. Our aim in this research is to monitor people's emotions during the COVID-19 pandemic. The intuition behind considering such tweets is that most of Arab people travel to Iran and there was an aware of transmission of the virus from Iran to the Arab world countries. The target is the Arabic language regardless of the country.
A better discussion is added after modifying the experiments on our dataset.
The conclusion is updated as well.
We almost change the way of annotation therefore most of the paper sections are modified.
Point 2: there are some errors in the reference list.
Response 2: All references issues resolved by following the template of the Information MDPI journal